# The Emerging Role of CD8^+^ T Cells in Shaping Treatment Outcomes of Patients with MDS and AML

**DOI:** 10.3390/cancers17050749

**Published:** 2025-02-22

**Authors:** Athanasios Tasis, Theodoros Spyropoulos, Ioannis Mitroulis

**Affiliations:** 1Translational Research and Laboratory Medicine Unit, First Department of Internal Medicine, University Hospital of Alexandroupolis, Democritus University of Thrace, 68100 Alexandroupolis, Greece; tasis.thanasis@gmail.com; 2Department of Hematology, University Hospital of Alexandroupolis, Democritus University of Thrace, 68100 Alexandroupolis, Greece; theodspiropoulos@gmail.com

**Keywords:** CD8^+^ T cells, myelodysplastic neoplasms (MDS), acute myeloid leukemia (AML), senescence, exhaustion, treatment response, allogenic hematopoietic stem cell transplantation (allo-HSCT), chemotherapy, hypomethylating agents (HMAs), immunotherapy

## Abstract

Cytotoxic immune cell populations, including CD8^+^ T cells, play a critical role in anti-tumor immunity, suppressing tumor progression. For this reason, recent therapeutic algorithms include immunotherapeutic agents that target such cell populations for the treatment of patients with a variety of solid tumors. In contrast to solid tumors, such immunotherapeutic strategies failed to show effectiveness in the treatment of patients with myeloid malignancies, including myelodysplastic neoplasms (MDS) and acute myeloid leukemia (AML). For this reason, the study of immunophenotypic and molecular changes in cytotoxic effector cell populations in patients with such disorders would provide the necessary evidence for the development of novel immunotherapies. In this review, we provide an overview of experimental studies describing the modulation of CD8^+^ T cell populations in patients with myeloid neoplasms, focusing on whether and how specific cell phenotypes are linked to disease progression and treatment.

## 1. Introduction

### 1.1. Brief Overview of CD8^+^ T Cells and Their Established Role in Immune Surveillance

CD8^+^ T lymphocytes are crucial components of adaptive immunity, playing a pivotal role in immune surveillance [1]. Immune surveillance is particularly important in the context of solid tumors, mainly due to the distinct antigenic profiles these malignancies often present [2]. Solid tumors frequently overexpress specific tumor-associated antigens (TAAs), which serve as prime targets for CD8^+^ T cells. The ability of cytotoxic T lymphocytes (CTLs) to recognize and target these TAAs is crucial in initiating effective anti-tumor responses [3,4,5]. Infiltration of solid tumors by CD8^+^ T cells, often paired with an enhanced interferon (IFN) signature, is associated with improved patient outcomes, compared to “non-T cell inflamed tumors”, strongly reflecting the significance of CTLs in the effective clearance of neoplastic cells [6,7,8,9,10]. However, the tumor microenvironment (TME) can severely influence the function of CTLs, hampering their anti-tumor response [6,7,11,12]. Cancer cells employ a wide range of immune evasion strategies, mainly through induction of immunosuppression and active disruption of CD8^+^ T cell functionality [13]. One of the most well-described mechanisms of immune evasion involves the overexpression of programmed cell death protein-ligand 1 (PD-L1) by tumor cells. PD-L1 binds to the programmed death receptor-1 (PD-1), expressed on CD8^+^ T cells, leading to T cell exhaustion, a state of reduced cytokine production and cytotoxic activity [14]. Hallmarks of T cell exhaustion include high expression of a wide range of inhibitory receptors, impaired cytotoxic function, and inability to produce key cytokines like IL-2, TNF, and IFN-γ [15,16]. In addition to PD-1 and PD-L1, other immune checkpoint molecules, such as TIM-3, TIGIT, and LAG-3, are overexpressed in the TME, further promoting T cell dysfunction (9, 27, 28). Abnormal expression of all these molecules in the TME contributes to the suppression of CTL activity and shields malignant cells from immune-mediated destruction [17,18,19]. Other mechanisms include hypoxia and general metabolic stress or physical barriers created by the extracellular matrix [20].

Altogether, these mechanisms highlight the dynamic interplay between CD8^+^ T cells and the TME, pointing out both the potential and the challenges of mounting an effective anti-tumor response by harnessing CD8^+^ T cells in cancer immunotherapy [21]. Over the years, the field of immunotherapy, which aims to restore T cell functionality against malignant cells, has made substantial advancements, showing remarkable promise in improving the clinical outcomes for patients with solid tumors [22,23]. Anti-cytotoxic T lymphocyte antigen 4 (Anti-CTLA-4) and anti-PD-1 antibodies, either as monotherapies or as combinational therapy, have been used for over a decade for the treatment of melanoma, resulting in substantial outcomes [24,25,26,27,28]. Furthermore, PD-1/PD-L1 blockade is currently used in the treatment of patients with solid tumors, either as monotherapy or in combination with other treatment regimens [29,30,31,32]. However, even with some remarkable results, not all patients benefit from immunotherapy, and the need for robust and easily applicable predictive biomarkers still remains [33].

### 1.2. Myelodysplastic Neoplasms (MDS) and Acute Myeloid Leukemia (AML)

MDS encompass a group of heterogenous clonal myeloid disorders primarily characterized by ineffective hematopoiesis, cytopenia, and increased incidence of progression to secondary AML [34,35]. Myeloid malignancies are generally characterized by broad molecular heterogeneity [36,37]. Accumulation of genetic abnormalities and epigenetic alterations at the hematopoietic stem cell level are the primary drivers of MDS [36,38]. Over time, the clonal expansion of dysplastic malignant cells leads to disease progression and subsequent AML transformation [39,40,41].

Despite the recent advances, treatment options for MDS and AML remain limited as they are confronted by significant challenges, mainly due to the profound heterogeneity of these malignancies. The treatment algorithm for these patients is tailored to the disease characteristics, patient condition, and their associated risk assessment [42,43]. Based on the revised international prognostic scoring system (IPSS-R), MDS patients are stratified into lower- and higher-risk groups [44]. Conversely, AML treatment is guided by the European LeukemiaNet (ELN) risk classification [43]. The treatment of lower-risk MDS (IPSS-R ≤ 3.5) remains mainly supportive and aims to improve quality of life. In more detail, the treatment plan involves transfusions, if needed, improvement of cytopenias by providing stimulating factors and combating anemia by using erythropoiesis-stimulating agents (ESAs), such as Erythropoietin or Luspatercept [45,46,47,48,49]. However, in the case of higher-risk MDS (IPSS-R > 3.5), the main goal is limiting the progression rate to AML and prolonging patient survival. To this date, treatment with the Hypomethylating agents (HMAs) Azacytidine (AZA) and Decitabine (DEC) remains the only available treatment options for elderly high-risk MDS patients [47]. HMAs do not eliminate the disease but rather prolong its natural progression course to AML without offering a curative option. In AML, treatment choices differ based on the patient’s status [50]. Younger fit patients receive intensive chemotherapy, usually followed by allogeneic hematopoietic stem cell transplantation, while older patients receive lower-intensity therapies [50]. These include HMAs either as monotherapy or in combination with Venetoclax (VEN), a BCL-2 inhibitor [43,50,51]. Despite the limited range of approved treatment options, ongoing investigations are currently exploring novel targeted agents for the management of MDS and AML. One promising approach involves the inhibition of mutated IDH proteins, which presents a potentially viable therapeutic option for these myeloid neoplasms [52,53,54]. IDH1 and IDH2 inhibitors (IDHi) act by hindering mutant IDH proteins and thereby disrupting the production of the oncogenic metabolite 2-hydroxyglutarate (2-HG) [52,55]. While these inhibitors have displayed high overall response rates, resistance to IDHi monotherapy remains a notable challenge [56,57]. To overcome this, combination strategies have been investigated, incorporating IDHi with intensive chemotherapy or HMAs, both of which have shown promising results [58,59]. Another emerging targeted approach focuses on inhibiting menin, a protein encoded by the tumor suppressor *MEN1* gene, which has been implicated in a plethora of malignancies [60,61]. Menin inhibitors disrupt its interaction with KMT2A, ultimately interrupting the promotion of downstream leukemogenic factors [60,61,62]. In preclinical investigations using AML cell lines, combinational treatment with menin and BCL-2 inhibitors has demonstrated encouraging results [62].

### 1.3. Tumor Microenvironment in Myeloid Neoplasms

The BM microenvironment has a central role in the pathogenesis of myeloid malignancies, such as MDS and AML [63,64]. Alterations within the bone marrow microenvironment lead to the dysfunction of the immune cell compartment, including T cells, which further enhances the dysregulation of normal hematopoiesis, while promoting immune evasion and the emergence of malignant clones [18,65,66]. Immune cells in MDS and AML often exhibit a highly suppressed phenotype, which is the result of a multifaceted network of dynamic interactions between neoplastic cells, the immune cell compartment, and the BM TME [67,68,69]. This immunosuppressive landscape is further augmented by dysregulation of a wide range of soluble factors with immunomodulatory features which further dampens the anti-leukemic response [66,70]. Soluble factors such as IL-1B, IL-6, and TNF-α have been shown to promote leukemic cell growth and resistance to therapy [71], while IL-2 and IL-10 along with TGF-β can also promote the survival of malignant cells [71,72,73,74].

T regulatory cells (Tregs) and myeloid-derived suppressor cells (MDSCs) play an important role in creating and maintaining an immunosuppressed BM microenvironment [75,76,77]. The accumulation of Tregs has been strongly correlated with impairment of CD8^+^ T cell infiltration and functionality, as well as poor prognosis in disorders beyond hematologic malignancies [78,79]. An increased proportion of circulating Tregs [80,81] as well as BM Tregs [82,83] has been consistently observed in patients with MDS compared to healthy controls. Similarly, studies have demonstrated the aberrant expansion and functionality of Tregs in the peripheral blood (PB) [84,85,86] and BM [86,87,88,89] of AML patients. Intriguingly, it has been shown that Tregs expand profoundly more in the BM of MDS and AML patients than in the peripheral compartment [82,86]. BM Tregs, in MDS and AML, exhibit distinct phenotypic and functional characteristics, compared to their peripheral counterparts, including an exacerbated suppressive activity, thus further intensifying the BM immunosuppressive milieu [82,86]. There is a correlation between Treg enrichment and disease progression in both MDS and AML, with their proportion increasing significantly at the time of relapse [83]. Conversely, a decreased frequency of Tregs has been linked to favorable treatment responses, highlighting their critical role in modulating disease outcomes [83,84,86]. Consistent with data from other disorders [90], enrichment of Treg proportion in MDS and AML has been strongly associated with diminished recruitment, functionality, and cytotoxic activity of CD8^+^ T cells [82,91,92].

MDSCs facilitate immunoescape through the overexpression of immunosuppressive cytokines, which disrupts the function of T, NK, and dendritic cells [93,94]. Significantly elevated frequency of BM MDSCs has been observed in MDS compared to normal controls [95,96], which is associated with inhibition of CD8^+^ T cell-mediated cytotoxicity via the STAT3-ARG1 signaling axis [96]. Similarly, expansion of circulating MDSCs has also been observed in AML patients, which has been correlated negatively with patient prognosis and CD8^+^ T cell activity [97,98]. Increased numbers of MDSCs lead to the secretion of IL-10 and TGF-β, which subsequently leads to the enrichment of Treg proliferation and function. This interplay between MDSCs and Tregs results in a synergistic feedback loop which leads to suppression of CD8^+^ T cells proliferation and function, and ultimately in malignant cell immune evasion and disease progression [94,95,99,100].

Since immune dysfunction is present in the TME in myeloid malignancies, immunotherapies have been used in patients with myeloid neoplasms, including MDS and AML [34,101], however with limited success and considerable concern about the high probability of adverse events [102,103,104,105,106,107]. Despite the efforts of treating MDS and AML utilizing immune checkpoint inhibitors (ICIs) either as a monotherapy or in combination with hypomethylating agents (HMAs), the standard of care for elderly high-risk MDS and AML patients, no significant outcomes have been observed [108,109,110,111]. ICIs have also been used in AML following relapse after allogeneic hematopoietic stem cell transplantation (allo-HSCT), with some promising results being observed but again with toxicities and adverse events occurring [112]. Though many clinical trials are ongoing [113,114,115] testing different combinations of therapies, there is no clear indication of immunotherapies.

## 2. Implication of CD8^+^ T Cells in MDS and AML

### 2.1. Dysregulated CD8^+^ T Cell Functionality and Clonality in MDS and AML

Aberrant CD8^+^ T cell functionality and clonality have emerged as key contributors to immune dysregulation in MDS and AML. Dysfunction of CD8^+^ T cells, a result of chronic antigen exposure, leads to cell exhaustion and reduced cytotoxicity [116,117]. The leukemic BM microenvironment of MDS and AML further dysregulates the anti-tumor response of CD8^+^ T cells through the expansion and aberrant functionality of Tregs and MDSCs [17,18,92,100]. Moreover, altered chemokine signaling, a common finding in AML and MDS, suppresses T cell activity while also inhibiting their infiltration into the BM TME, promoting malignant cell dominance [118,119].

Many studies provide evidence of abnormal transcriptomic and immunophenotypic signatures of CD8^+^ T cells in MDS and AML, which lead to compromised anti-tumor responses (Figure 1). Jinglian Tao et al. reported decreased secretion of IFN-γ by peripheral CD8^+^ T cells from MDS patients and an increased frequency of PB CD8^+^TIM-3^+^ T cells compared to healthy controls, which displayed significantly lower levels of perforin and granzyme B compared to their CD8^+^TIM-3^−^ counterparts [120]. Additionally, previous studies in murine models of AML highlighted the association between upregulation of the PD-1/PD-L1 axis and high frequency of CD8^+^ T cells co-expressing PD-1 and TIM-3 with impaired anti-leukemic responses and disease progression [121,122]. Moreover, increased frequency of circulating TIGIT^+^TOX^+^CD73^−^ CD8^+^ T Cells co-expressing PD-1 or CD39 has been observed in newly-diagnosed AML patients compared to healthy individuals, indicative of an exhausted phenotype [123]. These two subsets also exhibited significant downregulation of the differentiation-associated markers CD127 and TCF-1, suggestive of a state of irreversible exhaustion [123]. In line with these findings, remission following induction chemotherapy has been associated with a significant reduction in intracellular TOX expression in BM CD8^+^ T cells, coupled with upregulation of perforin and granzyme B, hinting at a revitalized cytotoxic function [124]. Consistent with these studies, Kong et al. showed that elevated frequency of circulating CD8^+^PD1^+^TIM-3^+^ T cells in AML patients is correlated with increased relapse rates after transplantation. This cell population was also characterized by impaired cytokine production capacity, including reduced expression of IFN-γ, further hinting at an exhausted phenotype [15,125,126].

Over the years, T cell dysfunction has been linked with exhaustion-related signatures. However, senescence, a process interconnected with exhaustion but differentiated by distinct mechanistic features, has also emerged as a crucial contributor to T cell impairment [127,128,129]. Persistent immune activation leads to the emergence of antigen-specific highly differentiated senescent CD8^+^ T cells [130,131], which are characterized by downregulation of co-stimulatory molecules, like CD27 and CD28, and overexpression of CD57 and KLRG1. Senescent CD8^+^ T cells are in a state of cell cycle arrest, unable to proliferate upon TCR stimulation, and exhibit limited lytic activity, maintaining, however, their ability to secrete cytokines [127,132,133,134]. Clonal expansion of CD57^+^CD28^−^ CD8^+^ T cells has been observed in MDS patients [135,136]. In this direction, Knaus et al. performed an in-depth phenotypic and functional assessment of the T cell landscape in AML patients at diagnosis and post-chemotherapy. In this study, the authors observed that CD8^+^ T cells from newly diagnosed AML patients diverged significantly from a healthy-like profile and displayed a mixed exhausted/senescent phenotype [137]. More specifically, the authors reported lower frequencies of naïve CD8^+^ T cells with a respective elevation of terminally differentiated effector CD8^+^ T cells in AML patients compared to healthy controls. An increased ratio of terminal effector to naïve CD8^+^ T cells has also been verified in other studies [138,139]. Further analysis revealed lower expression of CD27, CD28, and CD127 in AML CD8^+^ T cells, hinting at a senescent phenotype [137]. Additionally, compared to healthy controls, CD8^+^ T cells from AML patients exhibited significantly increased expression of CD57, indicative of senescence [134,140], as well as the exhaustion markers PD-1 and 2B4 [15,16]. Additionally, in vitro stimulation of CD8^+^ T cells from patients with AML, with or without the presence of autologous blasts, resulted in significantly higher upregulation of CD57 expression compared to stimulation of CD8^+^ T cells derived from healthy controls. These CD8^+^CD57^+^ T cells exhibited low expression of the proliferation marker Ki67, pointing to decreased proliferative capacity aligning with a senescent phenotype. Interestingly, co-culture of healthy CD8^+^ T cells with AML blasts also led to increased CD57^+^ expression; however, the degree of CD57 upregulation was notably lower compared to AML CD8^+^ T cells [137]. Functional assessment of CD8^+^ T cell-mediated cytotoxicity against primary leukemic blasts, following activation through an anti-CD33/CD3 bispecific T cell-engaging (BiTE) antibody construct, highlighted the impaired lytic activity of CD8^+^CD57^+^ T cells compared to their CD8^+^CD57^−^ counterparts [137], a finding also confirmed later by Rutella et al. [141]. These findings reinforce the notion that leukemic blasts induce an immunosuppressive tumor microenvironment which promotes a senescent CD8^+^ T cell phenotype, further compromising the anti-tumor response.

Transcriptomic analysis of CD8^+^ T cells verified these results at the gene expression level. Consistent with the immunophenotypic findings, AML patients displayed overexpression of senescent-related genes, including *B3GAT1* (encoding CD57) and *KLRG1*, as well as inhibitory receptors, including *CD160*, *CD244* (which encodes 2B4), *LAG3* and *TIGIT*. Furthermore, CD8^+^ T cells from AML patients exhibited downregulation of CD28, as well as genes related to T cell adhesion and migration [137].

This finding is in line with a previous study from Dieu et al., which demonstrated that while AML CD8^+^ T cells maintain the ability to form cell conjugates with autologous blasts, their ability to effectively form immunological synapses is impaired [142]. Furthermore, it was observed that circulating CD8^+^ T cells from AML patients overexpress a set of genes related to T cell activation; however, these genes diverged from a “healthy” T cell activation signature when the two sets of genes were compared [142]. Sand et al. also showed that CD57^+^ CTLs are characterized by limited cytotoxic activity, which was partially attributed to decreased expression of adhesion molecules, resulting in their impaired adhesion to target cells [143].

Lastly, one critical factor of CD8^+^ T cell dysfunction in MDS and AML involves alterations in the physiological profile of their TCR repertoire [144,145,146,147]. The TCR repertoire naturally declines in diversity with age, while malignant conditions, such as MDS and AML, can lead to abnormal clonal expansion [146,147,148]. TCR repertoire skewing is driven by the expansion of specific clones of CD8^+^ T cells reflecting the stimulation by distinct leukemic antigens [146]. This restricted repertoire undermines the competence of CD8^+^ T cell immunosurveillance to recognize a broader range of antigens, eventually leading to limited T cell function and reduced cytotoxicity [136,149]. Studies have shown that MDS and AML patients are characterized by higher clonality and a hyper-contracted TCR repertoire, characterized by lower diversity and enrichment compared to healthy controls [146,147,150,151,152]. Additionally, the clonal expansion intensity has been found significantly higher in BM CD8^+^ T cells compared to CD8^+^ T cells in the peripheral compartment, once again pointing to a highly dysfunctional and suppressive BM TME in myeloid malignancies [136]. A recent study utilizing scRNA seq on BM CD8^+^ T cells demonstrated gradual deterioration of clonal diversity in newly diagnosed AML patients followed by relapsed/refractory (R/R) AML patients, compared to healthy individuals [150], a finding also observed by Feng et al. [152]. In the same study, the profound amount of hyperexpanded BM CD8^+^ T cells of R/R AML patients was correlated with declined plasticity and limited cellular transitions [150]. A common finding between studies investigating the TCR landscape of MDS and AML patients is the fact that naïve CD8^+^ T cells demonstrate the highest frequency of unique clonotypes, whereas effector CD8^+^ T cells are characterized by the most contracted phenotype, presenting the lowest degree of distinct clonotypes [146,147,150]. This finding has also been observed in AML patients post-HSCT, with effector BM CD8^+^ T cells presenting the highest levels of skewing [88]. Furthermore, CD8^+^ T oligoclonal expansion has been linked with inferior treatment outcomes in these patients [150]. Lastly, CD8^+^ clonally expanded T cells exhibit altered phenotypes that impair their functionality, mainly characterized by the expression of inhibitory molecules and exhaustion [117,150].

Impairment of CD8^+^ T cells has been strongly correlated with increased leukemic transformation rate [125,153,154], while dysfunctional CTLs correlate with lower remission rates and reduced overall survival [139,155]. Overall, derangement of CD8^+^ T cell functionality contributes largely to disease progression in myeloid malignancies and is associated with unfavorable disease outcomes and poor OS [139,150].

### 2.2. Role of CD8^+^ T Cells in Allo-HSCT and Chemotherapy

To date, the only possible curative option for patients with MDS and AML is allo-HSCT; however, a high frequency of relapse remains, while the role of CD8^+^ T cells in influencing response rates following transplantation or preventing relapses is not yet sufficiently understood. Noviello et al. studied the expression of inhibitory receptors (IRs) on T cells and the TCR profile of patients with AML post-HSCT (Table 1) [156].

Patients with AML, particularly those that achieved complete remission (CR), exhibited an increased ratio of CD8/CD4 T cells compared to healthy controls, both in PB and BM, consistent with an enhanced recovery of CD8^+^ T cells after transplantation [156]. An increased proportion of BM Tregs was observed in patients who relapsed (REL) compared to responders and healthy controls, while no difference was observed in the frequency of PB Tregs between the groups [156]. Notably, immunophenotypic analysis of the T cell compartment revealed differences associated with outcome exclusively in patients that received HLA-matched HSCT [156]. In contrast, no significant disparities were reported regarding the expression of IRs between CR and REL patients who received HLA-haploidentical HSCT, thus the authors shifted their focus to the HLA-matched cohort [156]. In this group, REL was associated with an accumulation of exhausted CD8^+^ T cells in the BM, as revealed by the significantly elevated expression of PD-1, TIM-3, and CTLA-4 on their BM CD8^+^ T cells compared to CR [156]. Unsupervised analysis showed increased co-expression of multiple IRs, including PD-1, KLRG1, and TIM-3, only in the case of REL patients compared to CR and healthy controls, who displayed significantly lower IR expression [156]. Further analysis revealed that the IR expression varied between subsets of CD8^+^ T cells based on their differentiation stage [156]. While no differences were observed in the IR profile of late differentiated CD8^+^ T cells between the groups, REL patients exhibited a highly exhausted profile in their early-differentiated BM CD8^+^ T cells [156]. Additionally, polyfunctional analysis of BM CD8^+^ T cells revealed that effector functions are impaired in REL patients, regardless of their differentiation stage, as demonstrated by their reduced degranulation and cytokine-secretion capacity [156]. Immunophenotypic analysis of BM samples collected at earlier post-HSCT timepoints showed that a robust exhausted phenotype is present in early differentiated BM CD8^+^ T cells only in patients who would eventually relapse, suggesting that a small subset of BM CD8^+^ T cells becomes functionally impaired early on in the post-HSCT setting and this phenomenon is linked with subsequent relapses [156].

Mathioudaki et al. recently dissected the single-cell transcriptomic landscape of BM CD34^+^ and T cells in AML samples 100 days following HSCT, revealing specific CD8^+^ T cell gene signatures associated with transplantation outcome [157]. Trajectory analysis revealed that T cells from CR patients are more advanced in pseudotime compared to REL, with the former group showing an increased abundance of clusters that included effector and mature memory T cell subsets alongside a decreased frequency of Tregs [157]. Gene regulatory analysis, utilizing SCENIC [158], showed that CD8^+^ T cells from REL patients display increased activity of transcription factors (TFs) such as *REL*, *RELB*, *NFKB1,* and *NFKB2*, all of which are involved in TNF signaling. This molecular pathway has been previously implicated in the pathogenesis of MDS and AML by favoring malignant cell survival and expansion [159,160,161]. Conversely, CD8^+^ T cells from CR patients exhibited significantly higher activity of *TBX21*, also known as T-bet, a key regulator of CD8^+^ T cell activation and effector functionality [162]. Subsequent differential gene expression analysis of CD8^+^ T cells further validated these findings. REL patients exhibited enrichment of TNF signaling, while CR patients showed a transcriptomic signature strongly associated with immune cell activation and effective anti-tumor response, as well as upregulation of cytotoxic genes like *GZMB* and *CX3CR1*. Of note, CR patients also showed upregulation of *ADGRG1* (encoding GPR56), which was correlated with co-expression of genes involved in effector functionality and cytotoxicity, such as *PRF1*, *GZMB*, *GNLY* and *NKG7*. Moreover, BM CD8^+^ T cells with high *GPR56* expression showed no significant expression of exhaustion-related genes, which is indicative of a functional mediator of anti-tumor immunity [157]. These findings were further verified at the proteomic level by flow cytometry. CD8^+^GPR56^+^ T cells were shown to expand after antigen exposure, gradually increasing in serial BM samples post-HSCT, while displaying a strong affinity in recognizing and exhibiting robust cytotoxic activity against malignant cells, whereas lower frequencies of this subset were observed in patients prone to eminent relapse [157].

Taken together, these findings highlight the rising role of CD8^+^ T cells in shaping the outcome of HSCT, with their functional and transcriptomic signatures contributing to transplantation outcomes and influencing disease relapses. Besides allo-HSCT, CD8^+^ T cells have also been described as critical mediators of immune responses during chemotherapy (Table 1, Figure 2). A previous gene expression analysis of PB CD8^+^ T cells, utilizing paired samples pre- and post-chemotherapy by Knaus et al. [137], has revealed divergent transcriptomic signatures between patients who achieved CR and non-responders (NR). Notably, no significant difference was observed in the gene expression profile between the two response groups at baseline [137]. However, in the post-treatment analysis, CR patients displayed a reversion to a healthy-like gene expression pattern, whereas NR displayed evidence of persistent immune dysfunction characterized by downregulation of costimulatory receptors, transcription factors, and cell adhesion genes, as well as upregulation of IRs and apoptosis-associated genes [137]. After chemotherapy, CR patients exhibited enrichment of chemokine signaling, costimulatory signaling, and a naïve-like CD8^+^ T cell signature [137]. On the other hand, NR showed enrichment of apoptosis and NFκB signaling, as well as upregulation of genes associated with exhaustion and senescence gene signatures [137]. Immunophenotypic analysis showed that NR patients maintained high levels of late differentiated CD8^+^ T cells throughout the treatment course, while CR patients demonstrated a significant reduction in this subset following chemotherapy [137]. Furthermore, CR patients also displayed a significant reduction in CD8^+^ T cells co-expressing multiple IRs post-chemotherapy, whereas NR showed increased expression of TIM-3 and PD-1 on their CD8^+^ T cells compared to their baseline levels [137].

Similar results were also reported by Tang et al. [138], showing that CR patients post-chemotherapy exhibited increased percentages of circulating Tn and Tcm cells, as well as restoration of CD28 expression, compared to their pretreatment levels. A comparison of the T cell architecture at baseline, based on treatment response, revealed that R/R patients had significantly lower levels of Tn and elevated proportions of CD8^+^PD1^+^ and CD8^+^CD57^+^CD28^−^ T cells compared to CR patients [138]. Additionally, increased frequency of the senescent CD8^+^CD57^+^CD28^−^ T cell subset was observed in patients with adverse prognostic factors, as defined per the ELN guidelines and correlated with inferior OS [138].

Mazziotta et al. recently expanded our understanding of the role of senescent CD8^+^ T cells in chemotherapy response by utilizing paired single-cell transcriptomics on longitudinal BM samples [139]. Initial bulk RNA-seq analysis of CD8^+^ T cells at baseline revealed enrichment of naïve and memory signatures in responders, which is consistent with prior works [137,138]. Moreover, the response to chemotherapy was linked to post-treatment upregulation of interferon-stimulated genes (ISGs), suggestive of enhanced activation of anti-tumor immunity in responders. Complementary flow cytometry analysis showed an increased frequency of BM CD8^+^CD57^+^ classified as terminally differentiated/senescent-like (Term/SenL) in non-responders compared to responders and healthy controls. Subsequent scRNA-seq analysis of BM T cells revealed transcriptomic profiles in CD8^+^ T cells associated with chemotherapy response. Interestingly, no definitive signature linked with exhaustion was detected in this study, which, despite contradicting previously reported works, aligns with other studies that have explored the transcriptomic landscape of AML at single-cell resolution [88,150,163]. In line with their immunophenotypic findings, the authors observed a higher abundance of the Term/SenL cluster in non-responders, whereas response to chemotherapy was associated with the presence of early memory CD8^+^ T cells (early Tm). In this direction, utilizing independent patient cohorts, increased Tm/Term ratio was correlated with favorable ELN risk stratification as well as better treatment response and overall survival. Trajectory analysis, utilizing Slingshot, delineated two distinct developmental CD8^+^ T cell fates, with Tn having the capacity to differentiate into either a functional active lineage or the Term/SenL subset, with the latter cell population being enriched in non-responders, post-chemotherapy. Consistent with earlier studies [88,146,150], TCR-seq revealed that CD8^+^ T cells bearing a terminal differentiated senescent signature displayed the highest degree of clonal expansion, while a skewed Term/SenL clonal expansion was linked to treatment resistance [139].

### 2.3. Role of CD8^+^ T Cells in HMA Treatment

Treatment with HMAs such as AZA and DEC [50,164,165] remains the long-lasting therapy approach for patients who are not eligible for chemotherapy and allo-HSCT. Even after twenty years of the implementation of HMAs against MDS and AML [166,167,168,169,170], the exact mechanism of action of these agents remains largely elusive [164]. Furthermore, the response rates are not satisfactory, with a substantial proportion of patients experiencing relapse [171]. Efforts to discover biomarkers predicting treatment response have not been sufficiently fruitful, further highlighting the complexity of their action [164].

Zhao et al. described that even though DEC does not significantly alter the proportion of circulating immune cell subsets, treatment with HMA was shown to induce IR expression on T cells and reduce the ability of CD8^+^ T cells to effectively produce cytokines (Table 2, Figure 2) [172].

In line with other studies [137,138,139,163], response to treatment was associated with higher abundance of Tn and reduced levels of terminal effector CD8^+^ T cells [172]. Notably, expression of exhaustion-related markers, including PD-1, on CD8^+^ T cells was independent of treatment outcome [172]. Functional analysis of CD8^+^ T cells, following in vitro stimulation, showed significantly elevated production of IFN-γ in responders versus non-responders, indicative of the important role of IFN-mediated anti-tumor immunity in these patients [172].

As previously mentioned, MDS and AML patients exhibit a highly contracted TCR profile [144,146]; however, evidence shows that TCR diversity and repertoire improve following AZA treatment [147]. Even though typical TCR repertoire metrics seem independent of treatment outcome, response to HMA has been associated with the emergence of novel CD8^+^ T cell clonotypes, while contracted clonotypes primarily persist in HMA non-responders [144,146,147]. Additionally, increased baseline TCR diversity and post-AZA T cell richness have been associated with improved event-free survival (EFS) and overall survival in AML patients [174].

Recently, we further investigated the phenotypic and transcriptomic profile of BM CD8^+^ T cells before the initiation of AZA treatment, aiming to uncover potential predictive biomarkers and novel treatment targets [163]. Using mass cytometry (CyTOF), we performed an in-depth immunophenotypic analysis on pre-treatment BM samples, revealing substantial qualitative and quantitative differences within T lymphocyte subsets amongst the patient groups [163]. The frequency of a terminal effector CD8^+^ T cell subset, characterized as CD57^+^CXCR3^+^CCR7^−^CD45RA^+^, was increased in AML compared to MDS, a finding that was further confirmed by flow cytometry in a secondary cohort of patients. Importantly, patients with MDS and AML who failed to respond to AZA exhibited increased pre-treatment levels of the CD8^+^CD57^+^CXCR3^+^ subset compared to responders, who instead displayed a higher baseline abundance of Tn cells [163]. Furthermore, a higher baseline frequency of CD8^+^CD57^+^CXCR3^+^ T cells was strongly predictive of poor overall survival, while no association was observed between the levels of this immune cell population and the mutational status of the patients [163]. We further engaged scRNA-seq to assess the transcriptional profile of BM-sorted CD8^+^ T cells from patients with MDS and secondary AML at the highest possible resolution, to identify molecular signatures in specific CD8^+^ T subpopulations tied to favorable outcomes [163]. scRNA-seq analysis did not identify any CD8^+^ T cell clusters characterized by a bona fide exhaustion signature, nor did we observe any significant differences in the expression pattern of exhaustion-related genes associated with treatment outcome [163]. This further reinforces the notion of alternative mechanisms impacting T cell functionality in myeloid malignancies, in contrast to solid tumors, in which T cell exhaustion dominates functional impairment [150,177]. Instead, differential expression analysis combined with TF regulatory analysis of the identified cytotoxic CD8^+^ T cell clusters revealed that treatment response is associated with enrichment of IFN signaling, which is in line with a study by Dey et al. [178] on MDS patients treated with AZA combined with anti-TIM-3 blockade. These observations collectively underscore the hypothesized IFN-mediated mechanism of action of AZA [179,180,181]. Conversely, AZA non-responders exhibited enrichment of TNF signaling, similar to the findings of Mathioudaki et al. [157], along with upregulation of genes involved in TGF-β signaling [163]. Enrichment of TGF-β signaling was associated with impaired cytotoxic signature compared to responders [163], in agreement with the established role of TGF-β in inhibiting CD8^+^ T cell-mediated cytotoxic function and promoting tumor evasion [73,182,183].

### 2.4. Role of CD8^+^ T Cells in HΜA and Venetoclax Combination Therapy

Treatment with AZA has been demonstrated to augment the recognition and elimination of malignant cells by CD8^+^ T cells [184]. This immunomodulatory effect is especially pronounced when AZA is combined with VEN [173]. This therapeutic strategy results in a synergistic effect, improving leukemic blast clearance compared to monotherapy. VEN increases the production of reactive oxygen species (ROS), which in turn induces the expression of T cell activation markers [173]. This ROS-induced T cell activation leads to enhanced cytotoxicity, specifically targeting malignant cells [173]. On the other hand, AZA exerts its effects, at least in part, through activation of the STING/cGAS pathway, inducing a viral mimicry state [173]. This phenomenon results in target cells becoming more vulnerable to CD8^+^ T cell-mediated clearance, further enhancing the venetoclax-augmented cytotoxic response [173].

The exact effect of VEN, either as a monotherapy or in combination with HMA, on CD8^+^ T cells is not yet fully understood. A CyTOF analysis of pre-treatment and post-VEN samples from AML patients treated with VEN monotherapy revealed that VEN alone induces minimal alterations in the frequencies of peripheral T cell subsets [185]. On the other hand, a recent study exploring the immunological effects of the HMA+VEN combination reported an increased proportion of CD8^+^ Tn cells, upregulation of TIM-3 expression on peripheral CD8^+^ T cells, and reduced co-expression of IRs like TIM-3 and PD-1 on CD8^+^ and CD4^+^ T cells following treatment, with the latter two findings identified to be VEN-specific effects [186]. Recent data suggest the potential of combining VEN with immunotherapeutic strategies, as a study employing a murine model of AML treated with a combination of VEN with anti-PD-1 therapy led to higher CD8^+^ T cell proliferation and enhanced blast clearance compared to the combination of AZA+VEN [187]. Another study reported that treatment of AML patients with HMA+VEN results in increased frequencies of circulating effector memory CD8^+^ T cells (Tem) and reduced IFN-γ production by peripheral CD8^+^ T cells. Moreover, resistance to HMA+VEN was associated with elevated production of IFN-γ by peripheral CD8^+^ T cells and increased expression of the exhaustion-related marker CD39 on CD8^+^ Tem cells [188]. In the same direction, Nagasaki et al. showed that in AML patients experiencing post-HSCT relapse, treatment with the combination of AZA+VEN, leads to upregulation of BCL-2 on CD8^+^ T cells, an increased frequency of cytotoxic CD8^+^ T cells with high granzyme B and IFN-γ expression, a lower proportion of exhausted PD-1^+^CD8^+^ T cells, and an increased frequency of BCL-2^high^ CD8^+^ Tem cells following treatment compared to baseline [189]. Of note, a recent study pointed out the potential incorporation of immunotherapeutic strategies, particularly T cell bispecific antibodies (TCBs) targeting the intracellular tumor antigen Wilms tumor 1 (WT1), into the AZA+VEN treatment regimen [190]. Utilizing an in vitro co-culture model of healthy donor-derived T cells and the OCI-AML3 cell line, the study showed that the addition of WT1-TCB to the combination of HMA+VEN results in enhanced T cell-mediated cytotoxicity and expansion, while reducing the production of pro-inflammatory cytokines [190]. These synergistic effects were further validated using T cells from AML patients and a humanized AML murine model, where this triple combinational therapy led to increased leukemic cell lysis, therefore supporting the rationale for integrating immunotherapy into AZA+VEN-based treatment approaches for AML [190].

### 2.5. Role of CD8^+^ T Cells in Immunotherapy

Even though immunotherapies, particularly ICIs targeting PD-1 and CTLA-4, have offered substantial clinical benefits in patients with solid tumors, comparable success is yet to be achieved in myeloid malignancies [106,112,191]. This highlights the significant gap in our understanding of T cell dysfunction in myeloid malignancies and further emphasizes the urgent need to identify effective strategies to restore CD8^+^ T cell functionality.

Penter et al. demonstrated the efficacy of CTLA-4 blockade post-HSCT is linked to an effective CD8^+^ T cell infiltration and activation [175]. CTLA-4 blockade was shown to systemically alter the proportion of T cell subpopulations and modify the phenotypic profile of CD8^+^ T cells, regardless of treatment outcome. Bulk RNA-seq of tumor-site biopsies, pre- and post-immunotherapy, did not reveal any significant changes in the transcriptomic profile of non-responders. In contrast, patients who achieved CR exhibited a gene expression profile indicative of a robust immune activation compared to baseline. This was evidenced by significant enrichment of pathways related to leukocyte and lymphocyte activation, as well as upregulation of genes associated with enhanced adaptive immune response. Furthermore, CR patients were characterized by increased CD8^+^ T cell infiltration in their biopsies, accompanied by increased levels of pro-inflammatory chemokines in their peripheral blood plasma, related to T cell activation and effector function, compared to baseline.

A recent study by Desai et al. [150], utilizing paired scRNA-seq and scTCR-seq on BM CD8^+^ T cells from healthy controls and AML patients, identified two unique effector CD8^+^ T cell subsets with distinct features between newly diagnosed (ND) and R/R patients with AML. Canonically exhausted BM CD8^+^ T cells, characterized by elevated co-expression of IRs, constituted only a very small percentage of the total CD8^+^ T cell population. R/R patients exhibited an increased abundance of CD8^+^ T cell clusters with a senescent-like signature compared to healthy controls and ND patients [150], once again highlighting the emerging role of senescence in the aberrant functionality of CD8^+^ T cells in these patients. Conversely, ND patients showed enrichment of effector CD8^+^ T cell clusters lacking a senescent-like signature [150]. Pseudotemporal analysis revealed that while effector CD8^+^ T cells from ND patients showed a continuous state of differentiation with a similar number of cells across pseudotime, almost the entire population of effector CD8^+^ T cells from R/R patients clustered in the last compartment of pseudotime, indicative of a preference for terminal differentiation in advanced stages of AML [150]. Furthermore, scTCR-seq analysis demonstrated that R/R patients exhibited the lowest TCR diversity amongst the groups, coupled with the highest degree of oligoclonal expansion [150]. Of note, the hyperexpanded cells from R/R patients overlapped significantly with the senescent-like identified CD8^+^ T cell clusters, further tying senescence and exacerbated clonal expansion to T cell impairment in refractory disease [150].

Another study exploring the secretome of T cells at the single-cell level following stimulation provided useful insights into the distinct polyfunctional landscape of T cells at baseline and its association with immunotherapy response [176]. Here, Root et al. noted a significant divergence between the polyfunctionality of BM and PB CD8^+^ T cells. PB CD8^+^ T cells showed no significant differences in their cytokine expression profile based on patient response, demonstrating the importance of investigating specifically the TME rather than the circulating immune compartment [176]. Importantly, at the baseline setting, a BM CD8^+^ T polyfunctional group expressing TGF-β was elevated in non-responders, while response to immunotherapy was associated with enrichment of a BM CD8^+^ T cell cluster expressing high levels of IFN-γ [176]. This observation strongly aligns with our findings regarding response to AZA in myeloid malignancies [163], further pointing out the importance of the balance between TGF-β and IFN-γ within the TME in shaping treatment outcomes across different treatment modalities (Table 2).

Chimeric antigen receptor (CAR) T cell therapy, a breakthrough in the field of immunotherapy, has also been successfully incorporated in various hematologic malignancies, particularly B cell lymphomas and multiple myeloma, demonstrating promising efficacy [192,193]. While ongoing investigations are currently exploring the potential utilization of CAR-T cells for MDS and AML treatment, studies so far have reported some conflicting data and substantial challenges related to safety and efficacy which still limit their wide use in clinical practice [105,194,195,196,197]. Studies and small-scale trials have explored the safety and efficacy of alternative CAR-T cell targets, including CD33, CD123, NKG2D, and CLL-1; however, concrete evidence supporting their clinical application has yet to be reported [105,198,199,200,201]. Despite the recent progress, the specific role of CD8^+^ T cells in CAR-T cell therapy still remains elusive. Driouk et al. demonstrated the efficacy of NKG2D-targeting CAR-T cells against AML cell lines and reported an upregulation of 41BB and other costimulatory molecules by CD8^+^ NKG2D CAR-T cells compared to control T cells [202]. Similarly, Sugita et al. reported that CAR-T cells targeting CD123, derived from CD8^+^ T cells, display efficient anti-AML activity, both against AML cell lines and primary AML blasts [203]. Co-culture experiments of these CAR-T cells and AML blasts revealed robust malignant blast cell lysis and enhanced production of activation cytokines. The efficacy was also investigated in murine AML models with satisfactory results, with a notable extension in overall survival [203].

## 3. Future Directions

A thorough investigation of the functional dynamics characterizing CD8^+^ T cells in these disorders remains an unmet necessity, as it holds the potential to elucidate their direct anti-leukemic responses and their ability to predict and dictate clinical outcomes across the diverse treatment approaches. A deeper understanding of how this crucial immune cell population modulates response outcomes to the available treatment approaches -including HMA, ICIs, chemotherapy, and allo-HSCT, as well as combinatory strategies- will provide invaluable insights. Deciphering the role of CD8^+^ T cells in shaping these therapeutic responses could pave the way for the development of more precise and long-overdue effective treatment strategies. Ideally, these strategies should be tailored to the unique immunogenomic profiles of individual patients or well-defined specific patient subsets, with the goal of improving the quality of life, clinical outcomes, and OS for patients with MDS and AML. Addressing this informational void requires robust patient recruitment efforts, coupled with the integration of cutting-edge technologies, including single-cell multi-omics to illuminate the dynamic and complex crosstalk between the BM TME and malignant cells. By unveiling this interplay, which likely governs the challenging anti-tumor immunity in myeloid malignancies, these efforts hold the potential to introduce novel and individualized therapeutic approaches for patients with MDS and AML.

## 4. Conclusions

Significant progress has been made towards unraveling the sophisticated mechanisms underlying myeloid malignancies, especially with the dawn of multi-omics and single-cell technologies [89,204,205,206]. Despite these advances, most of the studies focus on dissecting the genomic and epigenomic landscape of these disorders, as well as the intricate characteristics and dynamics of neoplastic cells, that primarily drive clonal evolution in MDS and AML [207,208,209,210,211,212]. This results in omitting the immune compartment, a critical parameter in determining clinical outcomes, particularly the dysfunction of CD8^+^ T cells, which serve as primary orchestrators of anti-tumor immunity.

Following the example of onco-immunology in the field of solid tumors, a similar approach could be transformative, if not revolutionary, for myeloid neoplasms. Employing single-cell multi-omics approaches to explore the overshadowed complex dynamics within the immune microenvironment has the potential to provide indispensable insights regarding patient prognosis and treatment responses. The insufficient results of immunotherapy in effectively treating these myeloid malignancies to date highlight the urgent need to address the gap of knowledge regarding the nuanced interactions between CD8^+^ T cells and neoplastic cells, and how they influence treatment outcomes.

## Figures and Tables

**Figure 1 cancers-17-00749-f001:**
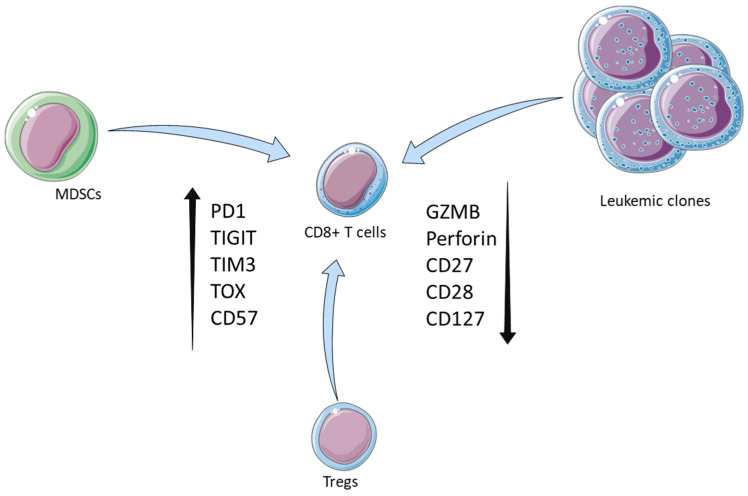
TME shapes CD8^+^ T cell function in myeloid neoplasms. Leukemic cells and immunosuppressive cell populations, such as MDSCs and Tregs, alter the phenotype and function of CD8^+^ T cells, resulting in an increased frequency of cells that express markers of exhaustion and senescence and in the decreased frequency of cells expressing naïve and cytotoxic cell markers.

**Figure 2 cancers-17-00749-f002:**
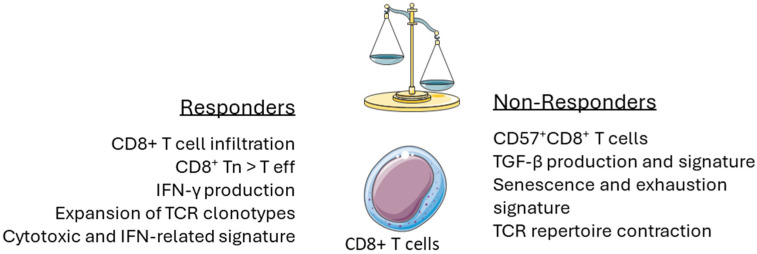
Main features of CD8+ T cell phenotype and molecular signature associated with response to treatment with chemotherapy or HMAs.

**Table 1 cancers-17-00749-t001:** CD8^+^ T cell correlatives of allo-HSCT and chemotherapy response.

Study	Treatment	Disease (Timepoint)	Main Findings
Noviello et al. [156]	allo-HSCT	AML (post HSCT)	Increased co-expression of multiple IRs by BM CD8^+^ T cells, particularly in their early differentiation stage, was associated with patient relapse. BM CD8^+^ T cells from relapsed patients displayed impaired effector functions compared to BM CD8^+^ T cells responders.
Mathioudaki et al. [157]	allo-HSCT	AML (post HSCT)	CR was associated with an enhanced cytotoxic signature and increased GPR56 expression by BM CD8^+^ T cells. Relapses were linked with an immunosuppressive milieu, including enrichment of TNF signaling.
Knaus et al. [137]	Chemotherapy	AML (pre- and post-chemotherapy)	Treatment failure was linked with enriched senescence and exhaustion signatures in CD8^+^ T cells. Conversely, responders displayed a reversion to a healthy-like transcriptomic profile, characterized by a less differentiated CD8^+^ T cell phenotype and increased expression of co-stimulatory molecules.
Tang et al. [138]	Chemotherapy	AML (pre- and post-chemotherapy)	Increased frequencies of CD8^+^ Tn and Tcm cells, along with lower proportions of terminally differentiated CD8^+^ T cells dictated chemotherapy success. In contrast, increased baseline proportion of the senescent-like CD8^+^CD57^+^CD28^−^ T cell subset was associated with poor OS.
Mazziotta et al. [139]	Chemotherapy	AML (pre- and post-chemotherapy)	Response to chemotherapy was associated with an enriched naïve and memory CD8^+^ T cell signature before treatment initiation and a significant overexpression of ISGs following chemotherapy. A reduced early CD8^+^ Tm/Term ratio was correlated with chemotherapy resistance and inferior OS.

**Table 2 cancers-17-00749-t002:** CD8^+^ T cell correlatives of HMA and immunotherapy response.

Study	Treatment	Disease (Timepoint)	Main Findings
Zhao et al. [172]	HMA	AML (pre- and post-HMA)	Responders demonstrated higher frequencies of CD8^+^ Tn and lower frequencies of CD8^+^ Teff cells. Moreover, CD8^+^ T cells from responders showed increased capacity of IFN-γ production upon stimulation.
Lee et al. [173]	Venetoclax and HMA	AML	Azacitidine sensitizes leukemic cells to T cell-mediated clearance, by triggering a viral mimicry response through the activation of the STING/cGAS pathway, while Venetoclax directly activates T cell-mediated response by enhancing their cytotoxicity.
Fozza et al. [147] Abbas et al. [144]	HMA	MDS (pre- and post-HMA)	Response to HMA was linked to a pronounced expansion of new TCR clonotypes, whereas treatment resistance was associated with a limited and restricted TCR repertoire.
Grimm et al. [174]	HMA	AML (pre- and post-HMA)	High baseline TCR diversity and a boost of T cell richness, 15 days post-AZA, was associated with treatment response and improved OS.
Tasis et al. [163]	HMA	MDS and AML (pre-HMA)	Increased baseline frequency of senescent-like BM CD8^+^CD57^+^CXCR3^+^ T cells was correlated with treatment failure and inferior OS. Response to AZA was linked with enhanced ISG and cytotoxic signatures prior to treatment initiation, whereas treatment failure was associated with enriched TGF-β signaling coupled with an impaired cytotoxic gene signature of BM CD8^+^ T cells.
Penter et al. [175]	Immunotherapy (CTLA-4 blockade)	R/R MDS/AML (post-HSCT)	Responders demonstrated substantial CD8^+^ T cell infiltration, alongside transcriptomic and phenotypic evidence of T cell activation and effector functionality.
Desai et al. [150]	Immunotherapy (PD-1 blockade)	AML (pre- and post-ICI)	R/R patients were characterized by a CD8^+^ T senescent-like signature. Treatment resistance was associated with TCR repertoire contraction.
Root et al. [176]	Immunotherapy (PD-1 blockade)	R/R AML (pre- and post-ICI)	Assessment of T cell polyfunctionality showed that response to immunotherapy was associated with increased baseline expression of IFN-γ and reduced expression of TGF-β by BM CD8^+^ T cells.

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
