# Peer review of "The Emerging Role of CD8+ T Cells in Shaping Treatment Outcomes of Patients with MDS and AML"

_cancers, 2025, doi:10.3390/cancers17050749_

Round 1
Reviewer 1 Report
Comments and Suggestions for Authors
The article describes the role of CD8+ T cells in treatment outcomes of patients with MDS AML in relation to chemotherapy, stem cell transplant, treatment with hypomethylating agents or checkpoints inhibitors.
The article is a comprehensive review of the topic, including published data from the same authors. the work is well written, and the discussion is well elaborated.
Minor comments:
(i) consider rephrasing the word sufficient on line 79, and adding the alpha symbol at line 123.
(ii) review the abbreviations for consistency, especially R/R, rel/refr., but also others.
Author Response
Comment 1: “consider rephrasing the word sufficient on line 79, and adding the alpha symbol at line 123”
Answer: We thank the reviewer for his/her comment. Corrected as indicated
Comment 2: “review the abbreviations for consistency, especially R/R, rel/refr., but also others”
Answer: Corrected as indicated
Reviewer 2 Report
Comments and Suggestions for Authors
This manuscript from Athanasios Thasis, Theodoros Spyropoulos and Ioannis Miltroulis provide a comprehensive review of the role of CD8+ T cells in MSD and AML, both at a cellular and molecular level but also providing interesting future points of practice. Overall, this manuscript is informative and easy to read to the non-specialist, however, several aspects deserve further consideration.
First: Conventional or typical cytotoxic T CD8+ cells, which produce perforin, granzyme B, IFN-gamma and TNF-alpha, seem to be the centre of this review. And that is adequate since these are the most prevalent tumor-infiltration lymphocytes in many cancers. However, alternate TCD8+ subsets with functions akin to TCD4+ cells have been identified, and these may have potential implications for prognosis and/or provide novel targets. For example, TGF-beta producing subsets in T CD8+ may represent CD8+ Treg cells and may be a mechanism for Azacytidine resistance in AML. Maybe explaining different functions of TCD8+ cells in a general way and indicating in the different sections wether the data come from bulk experiments or single cell experiments which can evaluate for these subsets would be an enhancement for this review.
Second, in the same spirit, the borders between immunosenescence and immune exhaustion are diffuse. This may confound the non-specialist reader, as this terms are somewhat mixed in the text. I would consider clarifying definitions in the introduction and keep consistency across the manuscript.
Third. Keeping with exhausted T CD8 cells, there is no information on the role of important factors as TOX.
Fourth: The section of iBCL + DNMT1 inhibitor/HMA (AZA or DEC) should be separated from the HMA and expanded, as it has enough entity and this represents standard therapy. Also information on novel drugs as IDH1/2 inhibitors or menin inhibitors, although scarce, is surely valuable.
Fifth: The manuscript would greatly benefit from a figure recapitulating major teaching points.
Author Response
Comment 1: “Conventional or typical cytotoxic T CD8+ cells, which produce perforin, granzyme B, IFN-gamma and TNF-alpha, seem to be the center of this review. And that is adequate since these are the most prevalent tumor-infiltration lymphocytes in many cancers. However, alternate T CD8+ subsets with functions akin to T CD4+ cells have been identified, and these may have potential implications for prognosis and/or provide novel targets. For example, TGF-beta producing subsets in T CD8+ may represent CD8+ Treg cells and may be a mechanism for Azacytidine resistance in AML. Maybe explaining different functions of T CD8+ cells in a general way and indicating in the different sections whether the data come from bulk experiments or single cell experiments which can evaluate for these subsets would be an enhancement for this review”
Answer: We thank the reviewer for his/her comment. Indeed, the manuscript is focused on conventional CD8+ T cells. The reviewer is right that other cell subsets, and especially TGF-beta producing cells, would be of potential significance. However, since to our knowledge their role in disease course in myeloid neoplasia is not established and the review is rather extensive, we focused on cytotoxic cell populations.
Comment 2: “in the same spirit, the borders between immunosenescence and immune exhaustion are diffuse. This may confound the non-specialist reader, as this terms are somewhat mixed in the text. I would consider clarifying definitions in the introduction and keep consistency across the manuscript”
Answer: We thank the reviewer for this comment. To address this, we have expanded the introduction to provide clearer definitions of immunosenescence and immune exhaustion, highlighting their distinctions. Additionally, we have ensured consistency in terminology throughout the manuscript to enhance clarity for non-specialist readers.
Comment 3: “Keeping with exhausted T CD8 cells, there is no information on the role of important factors as TOX”
Answer: In response to the reviewer's suggestion, we have incorporated information on key regulators of T cell exhaustion, including PD-1, CTLA-4, TIGIT, CD39, and TOX.
Comment 4: “The section of iBCL + DNMT1 inhibitor/HMA (AZA or DEC) should be separated from the HMA and expanded, as it has enough entity and this represents standard therapy. Also information on novel drugs as IDH1/2 inhibitors or menin inhibitors, although scarce, is surely valuable”
Answer: We appreciate the reviewer’s valuable insights. In response, we have separated the discussion of iBCL + DNMT1 inhibitor/HMA (AZA or DEC) from HMA monotherapy, creating a dedicated section titled “2.4. Role of CD8+ T Cells in HMA and Venetoclax Combination Therapy” to better highlight this treatment strategy. Additionally, we have incorporated information on IDH1/2 inhibitors and menin inhibitors into the introduction section of the manuscript, as recommended. However, our review of the literature did not reveal significant findings regarding the specific role of CD8+ T cells in response to treatment with IDH or menin inhibitors.
Comment 5: “The manuscript would greatly benefit from a figure recapitulating major teaching points”
Answer: We followed this comment, and we included a figure in the revised manuscript.
Reviewer 3 Report
Comments and Suggestions for Authors
I am grateful for the opportunity to review your valuable review article.
This review outlines the significance of CD8-positive T cells and their dysfunction in the pathogenesis of MDS and AML, and outlines the therapeutic potential of restoring CD8-positive T cell function in the treatment of these diseases and its clinical applications.
Regarding the first “simple summary”, we think it is unnecessary. Otherwise, it should be included in the “introduction” section.
The overall structure of the project seems to be well ordered and well organized. However, there are some parts that are difficult to understand with only descriptive information, so we encourage you to reconsider the explanation with some figures, which would make it even easier for readers to understand.
CAR-T cell therapy is a promising therapeutic area for the treatment of various malignant diseases, including MDS and AML. For the purpose of this review, it is necessary to mention CAR-T cells as an immuno-cell therapy as an independent section.
Author Response
Comment 1: “Regarding the first “simple summary”, we think it is unnecessary. Otherwise, it should be included in the “introduction” section”
Answer: According to the journals “Instructions for Authors” guidelines, the inclusion of a “Simple Summary” is a mandatory requirement for papers submitted to Cancers. Therefore, it cannot be removed or relocated to the introduction section.
Comment 2: “there are some parts that are difficult to understand with only descriptive information, we encourage you to reconsider the explanation with some figures, which would make it even easier for readers to understand”
Answer: We followed this comment, and we included a figure in the revised manuscript.
Comment 3: “CAR-T cell therapy is a promising therapeutic area for the treatment of various malignant diseases, including MDS and AML. For the purpose of this review, it is necessary to mention CAR-T cells as an immuno-cell therapy as an independent section”
Answer: Following the reviewer’s suggestion we have now incorporated information on CAR-T cells within the “Role of CD8 T cells in immunotherapy” section of our manuscript. We believe that this placement is the most appropriate, as it aligns with the broader discussion of CD8+ T cell involvement in immunotherapeutic strategies and the overall context of immune-cell therapies.
Round 2
Reviewer 3 Report
Comments and Suggestions for Authors
I am grateful for the opportunity to review your valuable revised article.
I am very satisfied that the authors were able to give serious consideration to my questions and suggestions, which were reflected in the content of the revised paper.